# Neurophysiological Evaluation of Neural Transmission in Brachial Plexus Motor Fibers with the Use of Magnetic versus Electrical Stimuli

**DOI:** 10.3390/s23084175

**Published:** 2023-04-21

**Authors:** Agnieszka Wiertel-Krawczuk, Juliusz Huber, Agnieszka Szymankiewicz-Szukała, Agnieszka Wincek

**Affiliations:** Department of Pathophysiology of Locomotor Organs, Poznań University of Medical Sciences, 28 Czerwca 1956 Street, No. 135/147, 61-545 Poznań, Poland

**Keywords:** brachial plexus, neural transmission, magnetic stimulation, electrical stimulation, cervical spinal roots, Erb’s point, neurophysiological diagnostics, motor-control assessment

## Abstract

The anatomical complexity of brachial plexus injury requires specialized in-depth diagnostics. The clinical examination should include clinical neurophysiology tests, especially with reference to the proximal part, with innovative devices used as sources of precise functional diagnostics. However, the principles and clinical usefulness of this technique are not fully described. The aim of this study was to reinvestigate the clinical usefulness of motor evoked potential (MEP) induced by a magnetic field applied over the vertebrae and at Erb’s point to assess the neural transmission of brachial plexus motor fibers. Seventy-five volunteer subjects were randomly chosen to participate in the research. The clinical studies included an evaluation of the upper extremity sensory perception in dermatomes C5–C8 based on von Frey’s tactile monofilament method, and proximal and distal muscle strength by Lovett’s scale. Finally, 42 healthy people met the inclusion criteria. Magnetic and electrical stimuli were applied to assess the motor function of the peripheral nerves of the upper extremity and magnetic stimulus was applied to study the neural transmission from the C5–C8 spinal roots. The parameters of compound muscle action potential (CMAP) recorded during electroneurography and MEP induced by magnetic stimulation were analyzed. Because the conduction parameters for the groups of women and men were comparable, the final statistical analysis covered 84 tests. The parameters of the potentials generated by electrical stimulus were comparable to those of the potentials induced by magnetic impulse at Erb’s point. The amplitude of the CMAP was significantly higher following electrical stimulation than that of the MEP following magnetic stimulation for all the examined nerves, in the range of 3–7%. The differences in the potential latency values evaluated in CMAP and MEP did not exceed 5%. The results show a significantly higher amplitude of potentials after stimulation of the cervical roots compared to potentials evoked at Erb’s point (C5, C6 level). At the C8 level, the amplitude was lower than the potentials evoked at Erb’s point, varying in the range of 9–16%. We conclude that magnetic field stimulation enables the recording of the supramaximal potential, similar to that evoked by an electric impulse, which is a novel result. Both types of excitation can be used interchangeably during an examination, which is essential for clinical application. Magnetic stimulation was painless in comparison with electrical stimulation according to the results of a pain visual analog scale (3 vs. 5.5 on average). MEP studies with advanced sensor technology allow evaluation of the proximal part of the peripheral motor pathway (between the cervical root level and Erb’s point, and via trunks of the brachial plexus to the target muscles) following the application of stimulus over the vertebrae.

## 1. Introduction

The anatomical complexity of the brachial plexus and its often multilevel damage require specialized in-depth diagnostics. The purpose is to select the appropriate treatment, assess its effectiveness, and provide prognostic information about its course [1,2,3,4,5]. Imaging of the brachial plexus, such as ultrasound or magnetic resonance imaging, provides important information about the nerve structures and surrounding tissues. Contemporary studies emphasize the importance of these tests, but they do not mention assessing the brachial plexus function [6,7,8]. Besides the clinical examination [9,10,11,12], the diagnostic standard for brachial plexus function should include clinical neurophysiology tests. Electroneurography (ENG) studies are used to assess the function of motor fibers and peripheral sensory nerves. Somatosensory evoked potentials are used to evaluate afferent sensory pathways. Needle electromyography analyses the bioelectrical activity of the muscles innervated by peripheral nerves originating from the brachial plexus. The results of the tests above determine the extent, type, and severity of the damage.

ENG of motor fibers uses a specific low-voltage electrical stimulus. It stimulates the nerve motor fibers, causing their depolarization, and the excitation spreads to the muscle, resulting in the generation of compound muscle action potential (CMAP). The strength of the electrical stimulus should be supramaximal, of sufficient intensity to generate CMAP with the highest amplitude and shortest latency. The CMAP amplitude reflects the number of conducting motor axons, and latency refers to the function of the myelin sheath and the rate of depolarization, mainly in fast-conducting axons [13]. Despite the advantages of this type of stimulation, it has limitations due to the physical properties of the electrical stimulus. The main limitation is the inability to penetrate through the bone structures surrounding the brachial plexus in its proximal part, at the level of the spinal roots, at the spinal nerves in the neck, and often at Erb’s point. Stimulation at Erb’s point may be complicated due to the individual anatomy of the examined person, such as obesity, extensive musculature, or past injuries at this level. This can significantly affect the CMAP parameters and give false positive results indicating pathology of the assessed motor fibers. In contrast to ENG, magnetic stimulus is used to induce motor evoked potential (MEP) [14,15]. Its use in brachial plexus diagnostics overcomes these limitations, which is of great clinical importance [16]. The propagation of excitation along the axon and elicitation of motor potential using a magnetic stimulus is similar to electrical stimulation. However, as some authors indicate, the applied magnetic stimulus may be submaximal [17,18,19,20] due to magnetic stream dispersion or insufficient power generated by the stimulation coil. Therefore, the assessment of MEP parameters may not reflect the actual number of excitable axons, and the interpretation of the results may incorrectly determine the functional status of the brachial plexus.

An MEP study can provide important information regarding the location of the injury, especially in cases of traumatic damage to the brachial plexus [2,3,18] where there may be multiple levels of impairment. The physical properties of the magnetic stimulus released from the generator device to penetrate bone structures would have to allow an assessment of the proximal part of the brachial plexus, especially at the level of the spinal roots. Scientific studies are mainly concerned with MEP efferent conduction studies in patients with disc–root conflict [21] and other neurological disorders [22,23]. Little attention has been paid to assessing the peripheral part of the lower motoneurone, including injuries of brachial plexus using MEP, studies of which may constitute the novum among the aims of the presented study. The main concern has been high-voltage electrical stimulation applied over the vertebrae [24,25]. To the best of our knowledge, apart from studies by Schmid et al. [26] and Cros et al. [27] from 1990, this paper is one of the few sources of reference values. Therefore, it makes a practical contribution to the routine neurophysiological diagnosis of brachial plexus injuries.

The aim of this study was to reinvestigate the hypothesis concerning the usefulness of the MEP test applied both over the vertebrae and at Erb’s point to assess the neural transmission of the brachial plexus motor fibers, with special attention to the functional evaluation of the short brachial plexus branches. The latter element has not been examined in detail [17]; most of the studies have been devoted to the evaluation of the long nerves, such as the median or ulnar. In addition, we formulated the following secondary goals: to compare the parameters of electrically evoked potentials (CMAP) with the parameters generated by magnetic stimulus (MEP), and to analyze whether these stimulation methods have compatible effectiveness and whether they could be used interchangeably during an examination. This would make it possible to select a method by taking into account the individual patient’s needs and the examination targets. Moreover, the additional aim of our work was to confirm that magnetic stimulation induces supramaximal potentials with the same parameters as during electrical stimulation, which was previously considered a methodological limitation [18]. A further study aim was to confirm an assumption that magnetic stimulation is less painful than electrical stimulation and better tolerated by patients during neurophysiological examinations, which has never before been examined.

## 2. Materials and Methods

### 2.1. Study Design, Participants, and Clinical Evaluation

Seventy-five volunteer subjects were randomly chosen to participate in the research. The ethical considerations of the study were compliant with the Declaration of Helsinki. Approval was granted by the Bioethical Committee of the University of Medical Sciences in Poznań, Poland (resolution no. 554/17). All the subjects signed a written consent form to voluntarily participate in the study without financial benefit. The consent included all the information necessary to understand the purpose of the study, the scope of the diagnostic procedures, and their characteristics. Before the study began, fifteen subjects declined to participate.

The subjects in the study group (N = 60) were enrolled based on the results of clinical studies performed independently by a clinical neurophysiologist and a neurologist. The exclusion criteria included craniocerebral, cervical spine, shoulder girdle, brachial plexus, or upper extremity injuries and other systemic disorders under treatment. The contraindications to undergoing neurophysiological tests were pregnancy, stroke, oncological disorder, epilepsy, metal implants in the head or spine, and implanted cardiac pacemaker or cochlear implant because of the use of magnetic stimulation. The results were analyzed blindly, satisfying intra-rater reliability. The medical history and clinical studies consisted of evaluating the sensory perception of the upper extremities according to the C5-C8 dermatomes and peripheral nerve sensory distribution, based on von Frey’s monofilament method [28]. The maximal strength of the upper extremity muscles was assessed using Lovett’s scale [29]. A bilateral clinical examination of each volunteer was performed once. Based on the clinical examination and medical history, the neurologist classified the subjects in the research group as healthy volunteers. After excluding 14 participants who did not meet the inclusion criteria and declining 4 others during the neurophysiological exams, the final group included 42 subjects. The characteristics of the study group (N = 42) and a flowchart of the diagnostic algorithm proposed in this study are presented in Figure 1 and Table 1. There were 40 right-handed participants and only 2 left-handed.

### 2.2. Neurophysiological Examination

All the participants were examined bilaterally once according to the same neurophysiological schedule. Each time, we used both magnetic and electrical stimuli to assess the function of the peripheral nerve and magnetic stimulus to evaluate neural transmission from the cervical spinal root. We applied stimulation three times at Erb’s point and at the selected level of the cervical segment, checking the repeatability of the evoked potential. The compound muscle action potentials (CMAP) recording during electroneurography (ENG) and motor evoked potential (MEP) induced by magnetic stimulation were analyzed. During the neurophysiological examination, the subjects were in a seated position, with relaxed muscles of the upper extremities and shoulder girdle, and in a quiet environment.

The KeyPoint Diagnostic System (Medtronic A/S, Skøvlunde, Denmark) was used for the MEP and CMAP recordings. External magnetic stimulus for the MEP studies was applied by a MagPro X100 magnetic stimulator (Medtronic A/S, Skøvlunde, Denmark) via a circular coil (C-100, 12 cm in diameter) (Figure 2A,B).

The strength of the magnetic field stream was 100% of the maximal stimulus output, which means 1.7 T for each pulse. The recordings were performed at an amplification of 20 mV/D and a time base of 5–8 ms/D. For the CMAP recording, a bipolar stimulation electrode and a single rectangular electric stimulus with a duration of 0.2 ms at 1 Hz frequency was used. The intensity of the electrical stimulus was 100 mA to evoke the supramaximal CMAP amplitude at Erb’s point. Such strength is obligatory and is determined by anatomical conditions and the fact that the nerve structures of the brachial plexus lie deep in the supraclavicular fossa. In the ENG studies, the time base was set to 5 ms/D, the sensitivity of recording to 2 mV/D, and 10 Hz upper and 10 kHz lower filters were used in the recorder amplifier. A bipolar stimulation electrode was used, the pools of which were moisturized with a saline solution (0.9% NaCl). The skin where the ground electrode and recording electrodes were placed was disinfected with a 70% alcohol solution; along with the conductive gel, this reduced the resistance between the skin and the recording sensors. The impedance did not exceed 5 kΩ.

In the ENG examination, the bipolar stimulation electrode was applied at Erb’s point over the supraclavicular region, along an anatomical passage of the brachial plexus motor fibers. If repetitive CMAP with the shortest latency and the highest amplitude was evoked at this point, the spot became the starting point for the application of magnetic stimulation at this level (hot spot). To assess the MEP from the spinal roots of the cervical segment, the magnetic coil was applied 0.5 cm laterally and slightly below the spinous process in accordance with the anatomical location of the spinal roots (C5–C8). In this way, the cervical roots were selectively stimulated.

For the recording of CMAP and MEP, standard disposable Ag/AgCl surface sensors with an active surface of 5 mm^2^ were used in the same location for both electrical and magnetic stimulus. The active electrode was placed over the muscle belly innervated by the peripheral nerve, taking the origin from the superior, middle, and inferior trunk of brachial plexus. The same selected muscles also represented a specific root domain in accordance with the innervation of the upper extremity through the cervical segment of the spine. The reference electrode was placed distal to the active ones, depending on the muscle, i.e., on the olecranon or the tendon [13]. A list of the tested muscles and their innervation (peripheral pathway and root domain), as well as the location of electrodes are given in Table 2.

The same parameters were analyzed for both the CMAP and MEP recordings. The amplitude of the negative deflection (from baseline to negative peak, measured in mV), distal latency (DL) (from visible stimulating artefact to negative deflection of potential, measured in ms), and standardized latency (SL) were calculated by the equation
SL = DL/LNS
where LNS is the length of the nerve segment between the stimulation point (Erb’s point) and the recording area on the muscle (measured in cm). A reliable value of standardized latency depends on an accurate distance measurement. Therefore, a pelvimeter, which reduces the risk of error in measuring the distance between the stimulation point and the recording electrode, was used in the research. This makes it possible to consider the anatomical curvature of the brachial plexus nerves. The standardized latency indicates a direct correlation between latency and distance. This is important in assessing the conduction of the brachial plexus short branches with regard to various anthropometric features of the examined subjects, such as the length of the upper extremities relative to height. In standard neurophysiological tests of short nerve branches, the F wave is not assessed, hence the calculation of the root conduction time for nerves such as axillary, musculocutaneous, etc., is not possible. In order to assess conduction in the proximal part of these nerves, the value of standardized latency was also calculated (proximal standardized latency, PSL) using the following equation:PSL = (MRL − MEL)/D
where MRL is the latency of MEP from the root level stimulation (measured in ms), MEL is the latency of MEP elicited from Erb’s point stimulation (measured in ms), and D is the distance between these two stimulation points (measured in cm). Therefore, the PSL value reflects the conduction between the cervical root and Erb’s point for each examined nerve. Distal latency and standardized latency correspond to speed conduction in the fastest axons. The amplitude of the recorded potentials and their morphology reflects the number of conducting motor fibers [13].

After undergoing neurophysiological tests, the subjects reported which of the applied stimuli (electrical or magnetic) evoked a painful sensation, as scored on a 10-point visual analogue scale (VAS) [30].

### 2.3. Statistical Analysis

The statistical data were analyzed using Statistica 13.3 software (StatSoft, Kraków, Poland) and are presented with descriptive statistics: minimal and maximal values (range), and mean and standard deviation (SD) for measurable values. The Shapiro–Wilk test was performed to assess the normality of distribution, and Leven’s test was used to define the homogeneity of variance in some cases. The results from the neurophysiological studies were compared to determine the differences between the sides (left and right), genders (female and male), stimulation techniques (electrical and magnetic), and stimulation areas (Erb’s point and cervical root). The changes in evoked the potential parameters between the groups of men and women were calculated with an independent Student’s *t*-test. In cases where the distribution was not normal, a Mann–Whitney U test was used. The dependent Student’s *t*-test (paired difference *t*-test) or Wilcoxon’s test (in the absence of distribution normality) was used to compare the differences between the stimulation methods, stimulation areas, and sides of the body. *p*-values less than 0.05 were considered statistically significant. The percentage of difference was expressed for each variable. An analysis of lateralization influence was not performed because there was only one left-handed volunteer. With regard to the results of the clinical tests, including pain measured by a 0–10 point visual analogue scale (VAS) and muscle strength measured by the 0–5 point Lovett’s scale, the minimum and maximum values (range) and mean and standard deviation (SD) are presented.

At the beginning of the pilot study, statistical software was used to determine the required sample size using the amplitudes from the MEP and ENG recordings with a power of 80% and a significance level of 0.05 (two-tailed) as the primary outcome variable. The mean and standard deviation (SD) were calculated using the data from the first 10 patients of each gender, and the software estimated that at least 20 patients were needed as a sample size for the purposes of this study.

## 3. Results

The research group was homogeneous in terms of age. We found statistically significant differences between the women and men concerning height, weight, and BMI (Table 1).

In the clinical study, the Lovett’s muscle strength score was found to be 5 on average for both men and women. This cumulative result applies to all assessed muscles bilaterally, i.e., deltoid, biceps brachii, triceps brachii, and abductor digiti minimi, and reflects the proper maximal muscle contraction against the applied resistance. The results of the sensory perception studies of the upper extremities, according to dermatomes C5–C8, were about the normal outcomes in the study group.

There were no significant differences in the CMAP and MEP between the right and left sides among women (N = 21) and men (N = 21). Hence, further comparative analysis of CMAP and MEP between the two groups refers to the cumulative number of tests performed (N = 42). The results are presented in Table 3.

The significantly prolonged latency of evoked potential in the men compared to the women is related to the greater distance between the stimulation point and the recording level, due to anthropometric features such as the length of the extremities, which are longer in men. However, this does not determine the value of standardized latency reflecting conduction in a particular segment. These values are comparable in the two groups for both types of stimulation (electrical and magnetic) and levels of stimulation (Erb’s point and cervical root) with generally no statistical differences. The exception is the C5 spinal root and Erb’s point stimulation (both electrical and magnetic) for the radial nerve. In the cases above, the standardized latency was significantly longer in the group of men. However, the percentage difference is only 8–11% and the numerical difference is only about 0.02 ms/cm, and these differences are not clinically significant. Similarly, there were significant differences in the amplitude of evoked potentials between women and men. In the assessment of the musculocutaneous nerve, CMAP and MEP generated from Erb’s point showed higher values in the men, while those generated from the ulnar nerve had higher values in the women. The difference is also between 10 and 16%, without clinical significance, and may have resulted from a measurement error, such as the cursor setting during the analysis of potentials.

Because the conduction parameters in the groups of women and men were comparable, further statistical analysis was conducted on 84 tests (both groups were combined). The parameters of potentials generated by electrical stimulus (CMAP) were compared with those of potentials generated by magnetic impulse (MEP). Stimulation in both cases was applied at Erb’s point. The data are presented in Table 4 and Figure 3.

The amplitude of CMAP was significantly higher after electrical stimulation than MEP after magnetic stimulation for all the examined nerves, in the range of 3–7%. This may have been due to the wider dispersion of electrical stimulation according to the rule of electrical field spread. The latency of the evoked potentials was significantly shorter after magnetic stimulation, which is related to the shorter standardized latency. Note that the difference in potential latency values using the two types of stimulation did not exceed 5%. This may be a result of the deeper and more selective penetration of magnetic impulses into tissues (based on the rule of magnetic field spread) and through the bone structures, and, thus, faster depolarization of the brachial plexus fibers. Figure 4 presents examples of CMAP and MEP recordings following electrical and magnetic stimulation at Erb’s point. The repeatability of the morphology of potentials with the use of both types of excitation is noteworthy.

The brachial plexus trunks are stimulated at Erb’s point in the supraclavicular area. In the area over the vertebrae, the spinous processes of the vertebrae are points of reference for the corresponding spinal root locations. In the cervical spine, according to the anatomical structure, the spinal roots emerge from the spinal cord above the corresponding numbered vertebrae. Figure 2A,B presents magnetic coil placements during the MEP study, while Table 5 gives data results.

The results show significantly higher amplitudes of the potentials after stimulation of the cervical roots compared to the potentials evoked at Erb’s point for C5 and C6. In the case of C8, the amplitude was lower than the potentials evoked at Erb’s point. It should be noted, however, that these values varied in the range of 9–16%, which, as explained above, is not clinically relevant. We also note the comparable values of proximal standardized latency (PSL) in the cervical root–Erb’s point segment for all the stimulated nerves. Figure 5 presents the MEP recordings after magnetic stimulation of the C5 to C8 cervical spinal roots.

The MEPs recorded from the cervical roots have a repetitive and symmetrical morphology. The MEPs have a lower amplitude at the C8 level than in the other studied segments (see Figure 5 and Table 5).

After undergoing neurophysiological tests, the subjects indicated the degree of pain sensation during stimulation according to a 10-point visual analogue scale (VAS) (see Table 6). The results indicate that they felt more pain or discomfort during electrical stimulation. The subjects described it as a burning sensation. They also indicated that magnetic stimulation was perceptible as the feeling of being hit, causing a more highly expressed motor action (contraction of the muscle as the effector of the stimulated nerve).

## 4. Discussion

Neuroimaging and basic clinical examinations of sensory perception and muscle strength are still the primary approaches for evaluating brachial plexus injury symptoms [31,32]. Neurophysiological diagnostics is considered supplementary, with the aim of confirming the results of the clinical evaluation.

The main novelty of the present study is that it proves the similar importance of magnetic and peripheral electrical stimulation over the vertebrae in evaluating the functional status of brachial plexus motor fiber transmission. The pros of our research are the neurophysiological assessment of the function of brachial plexus short branches, which are part of its trunks. Our studies prove the similarity of results obtained with the two mentioned methods following the excitation of nerve structures at Erb’s point. The latency and amplitude values of the potentials (CMAP, MEP) evoked at this level by two types of stimuli differed in the range of 2–7%. In routine diagnostic tests, this range of difference would not significantly affect the interpretation of the results of neurophysiological tests. Hence, we conclude that magnetic and electrical stimuli could be used interchangeably during an examination. We also proved that the range of excitation of motor fibers by a magnetic impulse may be supramaximal due to the stable and comparable MEP and CMAP amplitudes. The properties of supramaximal motor potential with the shortest latency were, in previous studies, attributed to the effects of electrical stimulation, which is commonly used in neurophysiological research. Many authors pointed to the limited diagnostic possibilities of the magnetic stimulus [31], the pros of which were examined in detail in this paper. This is crucial because of the different anthropometric features of patients and the possible extent of damage to the structures surrounding the brachial plexus. Past fractures, swelling, or post-surgical conditions at this level may limit the excitation of axons by electrical stimulus. The benefit of magnetic-induced MEP is that it is less invasive than electrical stimulation, as concluded from the VAS pain scores (see Table 6). The movement artifact associated with the magnetic stimulation may influence the quality of the MEPs recording, which should be considered during the interpretation of the diagnostic test results [32,33].

MEP studies allow evaluation of the proximal part of the peripheral motor pathway, between the cervical roots, contrary to low-voltage electrical stimulation. The comparable amplitudes of MEPs induced by magnetic stimulus recorded over the vertebrae with those recorded at Erb’s point, as shown in our study, could be the basis for the diagnosis of a conduction block in the area between the spinal root and Erb’s point. By definition, in a neurophysiological examination, conduction block is considered to have occurred when the amplitude of the proximal potential is reduced by 50% relative to the distal potential. In the opinion of Öge et al. [18], the amplitude of evoked potentials induced by stimulation of the cervical roots compared with potentials recorded distally using electrical stimulation may help to reveal a possible conduction block at this level. According to Matsumoto et al. [21], the constant latency of MEP induced by magnetic stimulation of the cervical roots was comparable with potentials induced by high-voltage electrical stimulation. In our opinion, similar to the method mentioned above, combining two research techniques using magnetic stimulation of the cervical roots or Erb’s point and conventional peripheral electrical stimulation is valid for neurophysiological assessment of the brachial plexus.

Previous studies on a similar topic by Cros et al. [27] involving healthy subjects revealed parameters of MEPs recorded from proximal and distal muscles of the upper extremities with the best “hot spots” from C4–C6 during stimulation over the vertebrae. They found that the root potentials were characterized by similar latencies, while the amplitudes recorded from the abductor digiti minimi muscle were the lowest following excitation at the C6 neuromere, contrary to our study, in which they were evoked the most effectively but with the smallest amplitudes following stimulation at C8 (see Table 5). We similarly recorded the largest amplitudes for MEPs evoked from the proximal muscles of the upper extremity. However, our study only involved magnetic stimulation over the vertebrae and not electrical stimulation, which was considered painful. In another study by Schmid et al. [26], magnetic excitation over the vertebrae at C7-T1 evoked MEPs with smaller amplitudes from distal muscles than proximal muscles compared to high-voltage electrical stimulation applied to the same area. Similar to our study, for MEPs following magnetic versus low-voltage electrical stimulation at Erb’s point, latencies were shorter and amplitudes were smaller, and the morphology was the same (see Table 2 and Figure 3). The standardized latencies were comparable for both types of stimulation, which was not reported by Schmid et al. [26].

In our opinion, when interpreting the results of neurophysiological tests of the brachial plexus, the reference values show a trend in terms of whether the parameters of the recorded potentials are within the normal range or indicate pathology [34]. When interpreting the results, special consideration should be given to comparing them with the asymptomatic side, which is the reference for the recorded outcome on the damaged side [35].

The results of the present study can be directly transferred to the clinical neurophysiology practice, due to the possibility of using two different stimuli in diagnostics to evoke the potentials with the same parameters that are recorded by non-invasive surface sensors. Magnetic stimulation appears to be less painful due to the non-excitation of the afferent component, contrary to electrical stimulation, where antidromically excited nociceptive fibers may be involved [36].

One of the study limitations that may have influenced the results, especially the parameters of latencies of potentials, was the anthropometric differences between women and men included in the study group. However, the gender proportions were equal, making the whole population of participants typical for European countries. Considering the number of participants examined in this study, it should be mentioned that due to comparable conduction parameters in the groups of women and men, the final statistical analysis covered 84 tests to compare the parameters of potentials evoked with electrical or magnetic impulses. Moreover, as mentioned in Section 2.3, at the beginning of the pilot study, statistical software was used to determine the required sample size, and it was estimated that at least 20 patients were needed for the purposes of this study.

## 5. Conclusions

This study reveals that the parameters of evoked potentials in CMAP and MEP recordings from the same muscles after the application of magnetic and electrical stimuli applied to the nerves of the brachial plexus are comparable. Magnetic field stimulation is an adequate technique that enables the recording of supramaximal potential (instead of the submaximal, which was reported in other studies [18]), which is the result of stimulation of the entire axonal pool of the tested motor path, similar to testing with an electric stimulus.

We found that the two types of stimulation can be used interchangeably during an examination, depending on the diagnostic protocol for the individual patient, and the parameters of evoked potentials can be compared. Moreover, in the case of patients sensitive to stimulation with an electric field, which is considered to cause pain in neurophysiological diagnostics, it is crucial to have the possibility of changing the type of stimulus. Magnetic stimulus is painless in comparison with electrical stimulus. We can conclude that the use of magnetic stimulation makes it possible to eliminate diagnostic limitations resulting from individual anatomical conditions or anthropometric features (such as large muscle mass or obesity). MEP studies allow us to evaluate the proximal part of the peripheral motor pathway (between the cervical root level and Erb’s point, and via trunks of the brachial plexus to the target muscles) following the application of stimulus over the vertebrae, which is the main clinical advantage of this study. It may be of particular importance in the case of damage to the proximal part of the brachial plexus. As a study of brachial plexus function, MEP should be compared to imaging studies in order to obtain full data on the patient’s functional and structural status.

## Figures and Tables

**Figure 1 sensors-23-04175-f001:**
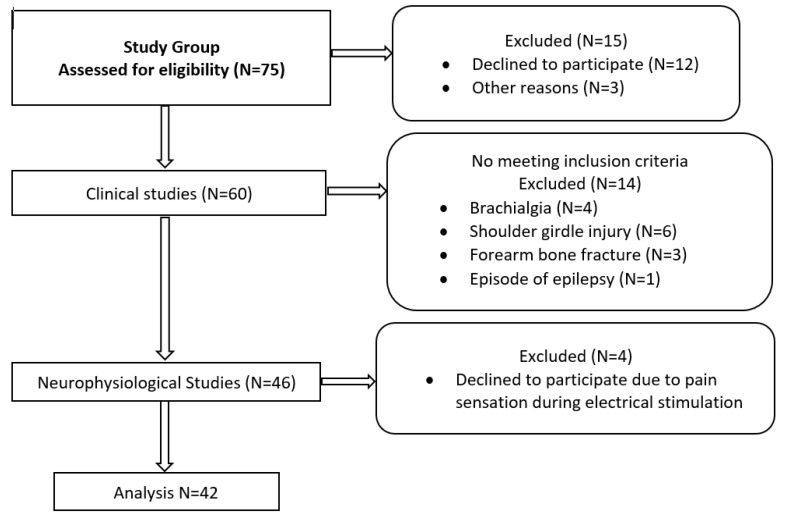
Flowchart of clinical and neurophysiological study algorithm.

**Figure 2 sensors-23-04175-f002:**
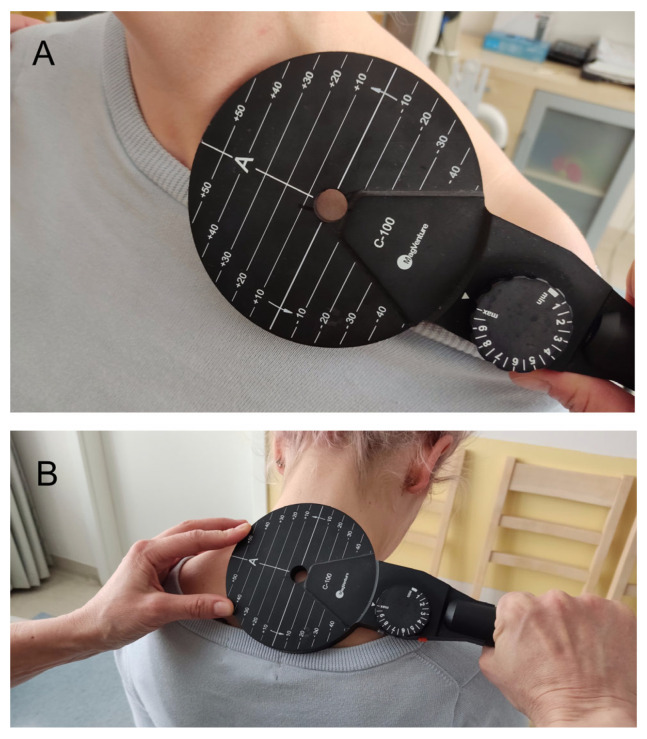
Photographs showing magnetic coil placement during MEP studies: (**A**) Erb’s point (level of brachial plexus trunks) stimulation and (**B**) stimulation over the vertebrae (level of cervical roots).

**Figure 3 sensors-23-04175-f003:**
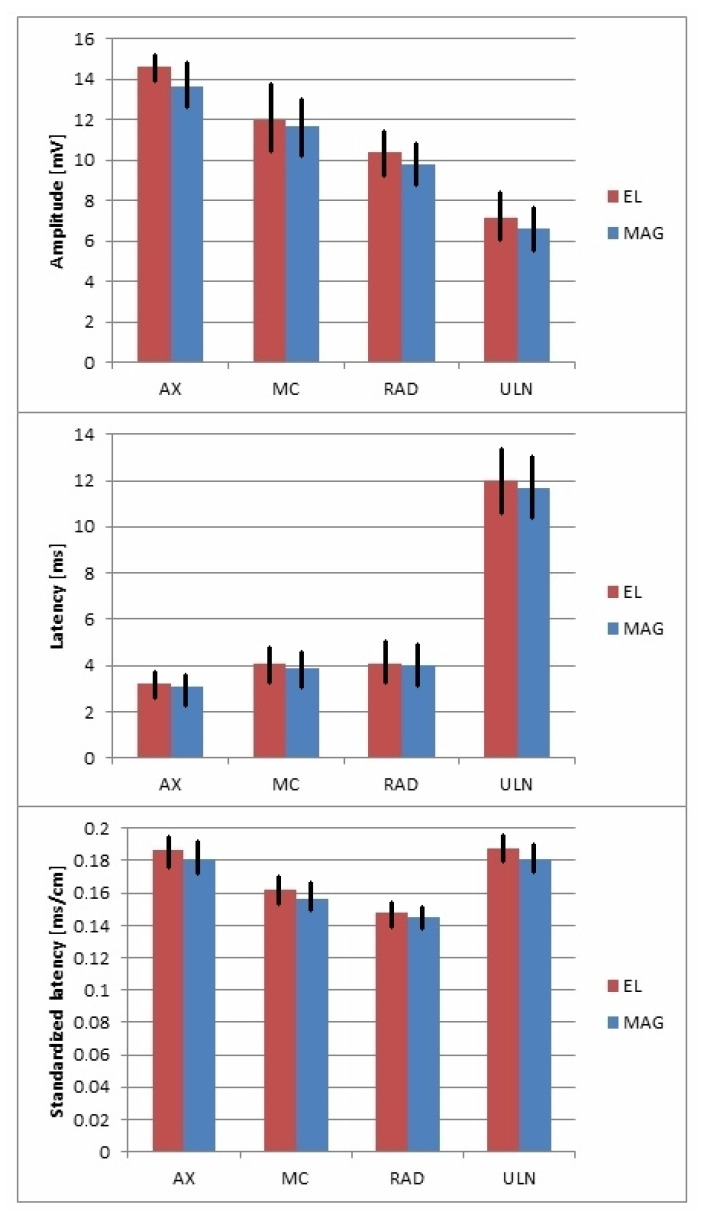
Graphical presentation of averaged CMAP and MEP recorded following electrical (EL) and magnetic (MAG) stimulation at Erb’s point, with standard deviations.

**Figure 4 sensors-23-04175-f004:**
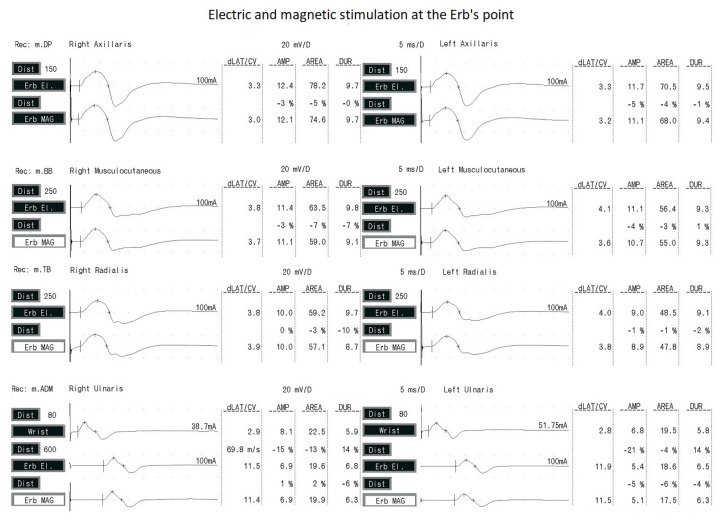
Examples of CMAP (upper traces) and MEP (lower traces) recordings after application of electrical and magnetic stimulation at Erb’s point on right and left sides. The distal point of ulnar nerve on the wrist was also stimulated (with electrical impulses only).

**Figure 5 sensors-23-04175-f005:**
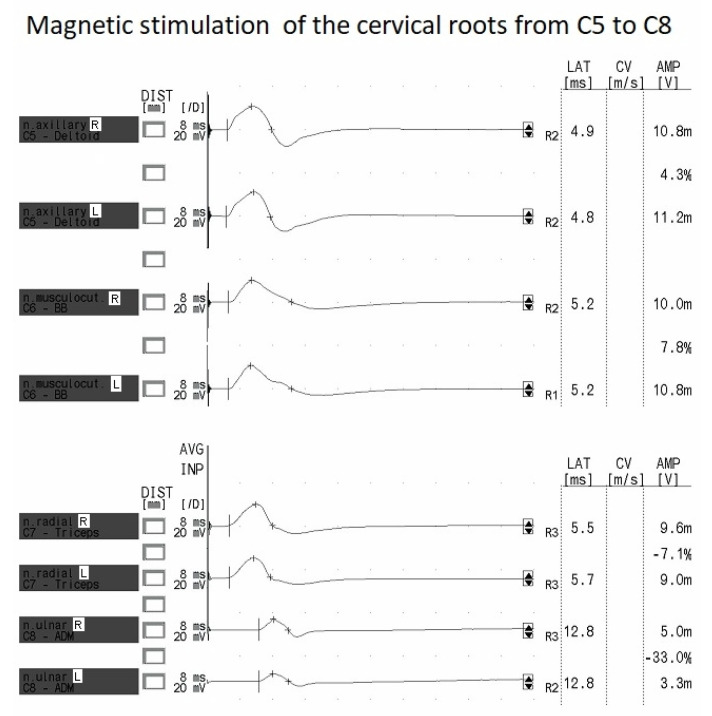
MEP recordings following magnetic stimulation of C5 to C8 cervical spinal roots.

**Table 1 sensors-23-04175-t001:** Data on demographic and anthropometric characteristics of subjects included in research group. Ranges, mean values, and standard deviations are presented.

Group	Age (Years)	Height (cm)	Weight (kg)	BMI (kg/m^2^)
N = 42	19–49 29.5 ± 9.2	155.0–190.0 172.3 ± 9.4	44.5–115.0 69.6 ± 15.3	17.6–32.1 23.2 ± 3.6
Female ♀ N = 21	20–47 31.3 ± 9.6	155.0–178.0 165.2 ± 7.3	44.5–72.1 58.1 ± 9.0	17.7–23.8 21.1 ± 2.5
Male ♂ N = 21	20–49 26.81 ± 6.3	170.0–190.6 179.4 ± 5.33	65.1–115.0 81.5 ± 13.2	21.2–32.1 25.2 ± 3.5
*p*-value (F vs. M)	0.384676 *	**0.0001**	**0.0002**	**0.007**

Notes: Significant differences are marked in bold. * Mann–Whitney U test; in other cases, independent Student’s *t*-test was used.

**Table 2 sensors-23-04175-t002:** Summary of examined muscles, their peripheral innervation, and root domain.

Muscle	Nerve	Brachial Plexus Trunk	Cervical Root (Significant Root Domain)
Deltoid (middle part) Active electrode—muscle belly Reference electrode—olecranon	Axillary nerve	Upper	C5–C6 (C5)
Biceps brachii Active electrode—muscle belly Reference electrode—olecranon	Musculocutaneous nerve	Upper	C5–C6 (C6)
Triceps brachii (long head) Active electrode—muscle belly Reference electrode—olecranon	Radial nerve	Upper, middle, lower	C6–C8 (C7)
Abductor digiti minimi Active electrode—muscle belly Reference electrode—muscle tendon	Ulnar nerve	Lower	C8–T1 (C8)

**Table 3 sensors-23-04175-t003:** Comparison of results of neurophysiological tests between women (*n* = 42) and men (*n* = 42) using electric and magnetic stimulation at Erb’s point and magnetic stimulation of cervical roots from C5 to C8. Ranges, means, and standard deviations are presented. ^†^ Significant *p*-values < 0.05.

Nerves/Method		Parameters											
		Amplitude (mV)			Latency (ms)			Standardized Latency (ms/cm)			Distance (cm)		
		Women	Men	*p*-Value (Difference)	Women	Men	*p*-Value (% of Change)	Women	Men	*p*-Value (% Change)	Women	Men	*p*-Value (% Change)
AX	EL	10.6–18.3 14.8 ± 2.0	10.1–16.6 14.1 ± 2.0	0.59 2.1%	2.4–3.8 3. 1 ± 0.12	3.3–4.5 3.4 ± 0.23	0.000002 *^†^ 12%	0.15–0.24 0.19 ± 0.013	0.14–0.22 0.16 ± 0.03	0.6 * 1.1%	15–18 15.4 ± 0.8	15–25 18.0 ± 2.3	0.0002 *^†^ 13%
	MAG	9.6–19.5 13.4 ± 2.1	10.4–16.1 13.5 ± 1.9	0.67 2%	2.6–3.3 2.96 ± 0.1	2.6–3.9 3.3 ± 0.25	0.00004 ^†^ 10%	0.15–0.24 0.19 ± 0.01	0.14–0.22 0.19 ± 0.01	0.82 * 1%			
	C5	6.1–22.7 15.5 ± 4.1	8.2–23.3 13.2 ± 4.4	0.12 * 14%	4.1–4.8 4.45 ± 0.28	4.6–5.5 5.1 ± 0.23	0.0000002 *^†^ 12%	0.1–0.19 0.14 ± 0.03	0.14–0.3 0.16 ± 0.02	0.008 ^†^ 11%	9–13 10.5 ± 08	8–12 10.6 ± 1.0	0.45 * 1%
MC	EL	7.4–14.8 10.3 ± 2.4	8.3–17.0 13.1 ± 2.33	0.004 ^†^ 17%	3.1–5.4 3.83 ± 0.5	4.1–4.4 4.35 ± 0.22	0.00004 *^†^ 12%	0.13–0.20 0.17 ± 0.03	0.16–0.19 0.17 ± 0.009	0.22 * 4.3%	21–27 22.4 ± 1.6	25–31 26.1 ± 1.5	0.00001 *^†^ 13%
	MAG	7.1–14.2 10.5 ± 2.2	8.4–16.6 12.53 ± 2.22	0.02 ^†^ 14%	3.0–4.3 3.62 ± 0.33	3.7–4.9 4.12 ± 0.21	0.000003 ^†^ 13%	0.14–0.20 0.15 ± 0.03	0.14–0.18 0.14 ± 0.015	0.68 1.2%			
	C6	8.1–18.2 12.1 ± 3.3	7.7–21.3 13.1 ± 3.01	0.27 8,1%	4.1–6.1 5.2 ± 0.5	5.5–6.2 6.3 ± 0.22	0.000013 ^†^ 12%	0.12–0.25 0.17 ± 0.04	0.13–0.24 0.17 ± 0.02	0.68 2%	9–13 10.6 ± 1.1	11–13 11.2 ± 0.5	0.18 * 4.2%
RAD	EL	7.2–12.6 9.5 ± 1.3	6.6–13.4 10.8 ± 1.75	0.04 ^†^ 10%	3.1–4.8 3.6 ± 0.4	4.1–5.6 4.5 ± 0.5	0.000044 *^†^ 18%	0.13–0.18 0.15 ± 0.03	0.15–0.18 0.16 ± 0.02	0.011 *^†^ 7.1%	21–28 25.1 ± 1.42	25–36 28.7 ± 3.1	0.00005 *^†^ 13%
	MAG	7.0–12.6 9.4 ± 1.5	8.2–11.8 10.0 ± 1.2	0.14 6%	3.1–4.1 3.5 ± 0.45	3.5–5.1 4.43 ± 0.48	0.000002 ^†^ 18%	0.13–0.16 0.14 ± 0.02	0.14–0.18 0.16 ± 0.02	0.003 *^†^ 8.2%			
	C7	4.3–18.7 10.1 ± 3.9	5.5–17.4 11.4 ± 3.31	0.42 8.1%	4.3–6.1 5.2 ± 0.61	5.5–7.6 6.4 ± 0.2	0.000005 ^†^ 16%	0.14–0.25 0.17 ± 0.04	0.13–0.25 0.17 ± 0.05	0.8 1.1%	9–13 10.5 ± 0.7	10–13 11.4 ± 0.5	0.002 *^†^ 8.3%
ULN	EL	5.2–10.6 7.4 ± 1.3	5.0–8.7 6.1 ± 1.03	0.013 ^†^ 12.9%	9.8–13.3 11.5 ± 0.8	11.4–14.5 12.6 ± 0.8	0.00028 ^†^ 8%	0.18–0.22 0.20 ± 0.02	0.18–0.23 0.20 ± 0.03	0.88 * 1.1%	56–68 60.4 ± 3.6	54–74 66.1 ± 5.12	0.001 *^†^ 9%
	MAG	5.4–9.6 7.2 ± 1.4	3.2–8.5 6.2 ± 1.25	0.012 ^†^ 14.8%	9.5–13.3 11.3 ± 0.1	10.5–13.3 12.4 ± 0.6	0.000074 ^†^ 9%	0.18–0.22 0.17 ± 0.03	0.17–0.22 0.19 ± 0.03	0.36 * 2%			
	C8	1.6–14.2 5.6 ± 3.1	1.0–9.7 5.4 ± 2.8	0.46 11%	11.6–14.5 13.1 ± 0.5	12.5–15.3 14.2 ± 0.5	0.000007 ^†^ 8.1%	0.13–0.26 0.14 ± 0.05	0.12–0.24 0.16 ± 0.05	0.49 * 4%	9–13 10.7 ± 1.1	10–14 11.5 ± 0.8	0.023 *^†^ 7.2%

Notes: *n*, number of tests; AX, axillary nerve; MC, musculocutaneous nerve; RAD, radial nerve; ULN, ulnar nerve; EL, electrical stimulation at Erb’s point; MAG, magnetic stimulation at Erb’s point; C5–C8, cervical roots. * Mann–Whitney U test; in other cases, independent Student’s *t*-test was used.

**Table 4 sensors-23-04175-t004:** Data of evoked potentials following application of two types of stimulation (magnetic and electrical) at Erb’s point (*n* = 84). Ranges, means, and standard deviations are presented. ^†^ Significant *p*-values at <0.05.

Nerves	Parameter								
	Amplitude (mV)			Latency (ms)			Standardized Latency (ms/cm)		
	EL	MAG	*p*-Value (% Change)	EL	MAG	*p*-Value (% Change)	EL	MAG	*p*-Value (% Change)
AX	10.3–18.7 14.5 ± 1.8	9.6–19.4 13.2 ± 2.0	0.0001 ^†^ (6.1%)	2.5–4.1 3.1 ± 0.28	2.5–3.8 3.0 ± 0.2	0.003 *^†^ (3.1%)	0.11–0.22 0.19 ± 0.02	0.14–0.22 0.18 ± 0.03	0.004 *^†^ (3.2%)
MC	7.6–17.0 122 ± 2.3	7.3–16.3 11.5 ± 2.2	0.02 ^†^ (3%)	3.0–5.5 4.2 ± 0.5	3.1–4.7 3.5 ± 0.3	0.002 ^†^ (5%)	0.12–0.18 0.16 ± 0.01	0.13–0.19 0.14 ± 0.02	0.002 *^†^ (4%)
RAD	6.6–13.1 10.3 ± 1.5	7.0–12.1 9.7 ± 1.4	0.003 ^†^ (6.3%)	3.0–5.8 4.3 ± 0.5	3.0–5.6 4.0 ± 0.6	0.004 ^†^ (2.2%)	0.13–0.18 0.13 ± 0.01	0.13–0.16 0.14 ± 0.01	0.02 *^†^ (2%)
ULN	5–10.9 7.0 ± 1.4	3.7–9.1 6.5 ± 1.4	0.0002 ^†^ (7.1%)	9.8–14.3 12.1 ± 1.0	9.7–13.8 11.8 ± 0.7	0.0001 ^†^ (3%)	0.17–0.21 0.19 ± 0.02	0.16–0.21 0.18 ± 0.02	0.0001 *^†^ (4%)

Notes: *n*, number of tests; AX, axillary nerve; MC, musculocutaneous nerve; RAD, radial nerve; ULN, ulnar nerve; EL, electrical stimulation at Erb’s point (nerve point of neck); MAG, magnetic stimulation at Erb’s point (nerve point of neck). * Wilcoxon’s test; in other cases, independent Student’s *t*-test was used.

**Table 5 sensors-23-04175-t005:** Comparison of MEP parameters for stimulation at Erb’s point and cervical root. Ranges, means, and standard deviations are presented. Significant *p*-values at <0.05 are marked with † symbol.

Nerve	Parameter					
	Amplitude (mV)			Latency (ms)		PSL (ms/cm)
	ERB	ROOT	*p*-Value (% Change)	ERB	ROOT	ROOT-ERB
AX/C5 (DP)	9.4–19.8 13.5 ± 2.3	6.0–23.8 14.2 ± 4.3	0.5 (3%)	2.5–3.8 3.0 ± 0.3	4.2–5.4 4.7 ± 0.6	0.1–0.2 0.17 ± 0.02
MC/C6 (BB)	7.2–16.3 11.6 ± 2.4	7.9–21.4 12.5 ± 3.1	0.01 † (9.2%)	3.0–4.9 3.9 ± 0.5	4.0–6.4 5.6 ± 0.5	0.11–0.24 0.16 ± 0.02
RAD/C7 (TB)	7.1–12.3 9.8 ± 1.2	4.7–18.6 11.3 ± 3.5	0.02† (11%)	3.1–5.3 4.0 ± 0.7	4.3–7.4 5.8 ± 0.8	0.12–0.24 0.15 ± 0.03
ULN/C8 (ADM)	3.4–9.53 6.5 ± 1.2	1.0–14.3 5.5 ± 3.5	0.01† (16%)	9.54–13.8 11.3 ± 0.9	11.7–15.1 13.4 ± 0.8	0.12–0.26 0.15 ± 0.03

AX, axillary nerve; DP, deltoid muscle; MC, musculocutaneous nerve; BB, biceps brachii muscle; RAD, radial nerve; TB, triceps brachii muscle; ULN, ulnar nerve; ADM, abductor digiti minimi muscle; PSL, proximal standardized latency. C5–C8: cervical spine root levels; ERB: magnetic stimulation at Erb’s point; ROOT: magnetic stimulation at root level.

**Table 6 sensors-23-04175-t006:** Results of mean values for 10-point visual analog pain scale (VAS) reported by women and men.

Group	VAS Score—MAG	VAS Score—EL
Women ♀ N = 21	3 (N = 15)	5 (N = 6)
Men ♂ N = 21	3 (N = 13)	6 (N = 8)

MAG, magnetic stimulation; EL, electrical stimulation.

## Data Availability

All the data generated or analyzed during this study are included in this published article.

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
