# Peer review of "Neurophysiological Evaluation of Neural Transmission in Brachial Plexus Motor Fibers with the Use of Magnetic versus Electrical Stimuli"

_sensors, 2023, doi:10.3390/s23084175_

Round 1

Reviewer 1 Report

The authors aimed to verify the hypothesis about the clinical usefulness of the motor evoked potential studies induced with the magnetic field (MEP) both applied overvebrtebrally and at Erb's point for the brachial plexus motor fibers neural transmission assessment. The study is meaningful. But I have several concerns before publication.

1.    The innovation of the paper is not clear.

2.    Electrical stimulation and magnetic stimulation are commonly used in clinical practice. Does this need to be verified?

3.    The sample is too small to draw reliable conclusions. I would like to recommend to add 20 for both male and female subjects.

4.    Please add standard deviation in figure 3.

5.    The discussion section needs to add more relevant literature to support its results.

6.    The conclusion needs to be rewritten.

Reviewer 2 Report

(1) The Abstract looks slightly longer and needs to shorten to fit the length of Sensors.   

(2) To satisfy the focus of the special issue in Sensors, it is suggested to revise the content by enhancing on the idea of “sensor” or “device” for neurophysiology. 

(3) There exists some typos and grammar mistakes in the article. Further proofreading is needed.

(4) Please check the correctness for the equation shown on Line 212. It seems to lack the bracket in the equation. 

(5) In Tables 3-5, it is suggested to replace the “Bold” format by specific symbol (such as †, * and so on) for those values of statistical significance. In addition, it is suggested to unify the significant p-values at “< 0.05” but not “≤ 0.05”. 

(6)   In Figure 5, the label of x axis is required. In addition, the corresponding traces for C5 to C8 should be marked or described in the context.  

Round 2

Reviewer 1 Report

The author's responses do not make me agree to "accept" the paper. Especially on the innovativeness and sample size of the article.
